# Spectral Characterization of Two-Photon Interference between Independent Sources

**Lifeng Duan, Aojie Xu and Yun Zhang \***

Department of Engineering Science, The University of Electro-Communications, 1-5-1 Chofugaoka, Chofu-shi, Tokyo 182-8585, Japan
\* Correspondence: zhang@ee.uec.ac.jp; Tel.: +81-042-443-5898

**Abstract:** We present a spectral characterization of two-photon nonclassical interference on a beam splitter (BS) between a weak coherent state source and another source, which emits a phase-randomized weak coherent state, a single-photon state, or a thermal state. Besides spectral characteristics, the average photon number ratio in a given time interval is also considered in our model. The two-photon coincidence probability of two outputs of the BS is numerically calculated with spectral bandwidth ratio and average photon number ratio. Furthermore, the noise of the detection system is taken into account. This also indicates that two-photon interference is able to significantly improve by subtracting two-photon contributions from the input state. All these parameters have a close relation to a real experiment performance and the results may pave new avenues for quantum information technology when two-photon interference between independent sources is necessary.

**Keywords:** two-photon interference; single-photon state; weak coherent state

## 1. Introduction

The interference between two single-photon wavepackets at a beam splitter (BS) is one of the critical effects in quantum optics. Coincidence events between the two single-photon counting detectors, which are placed in two output ports of BS, are investigated. From the view of quantum mechanics, there are no coincidence counts when two identical single photons feed the BS. The two-photon interference between two indistinguishable single photons was first observed employing photon pairs generated through spontaneous parametric down-conversion (SPDC), named the Hong–Ou–Mandel (HOM) dip [1]. The HOM dip can be understood as the destructive interference in quantum mechanics between the two probability amplitudes for single photons emerging in each of the two outputs. It can also be understood as the entanglement between two output fields, hence the interference is able to investigate using a second-order correlation function. Up to now, two-photon interference has also attracted considerable attention because of its involvement in fundamental quantum physics and potential applications in quantum information processing [2,3] and quantum key distribution [4–7].

In a quantum networking application, it becomes necessary to connect distant quantum devices on quantum nodes, such as quantum relays and repeaters [8–10]. A local quantum operation, such as the measurement of two-photon interference, can combine different pairs of entangled quantum bits. This operation allows qubits that never physically meet to become entangled. From the viewpoint of quantum networking applications, the two-photon interference between independent sources is the core technology. Until now, this operation has been performed using various kinds of single photon sources, such as nonlinear crystals [11,12], quantum dots [13], single atoms [14], or trapped ions [15,16]. In particular, the operation turned to observe two-photon interference between a single photon and a weak coherent source since it will significantly simplify the system for quantum networking [17–19]. Recently, it was demonstrated that two-photon interference effects can





be observed even between two weak coherent states. Although a HOM dip of 50% is restricted by the existence of the multi-photon probability of the Poisson distribution [20–26], the limitation can be overcome using a postselection technique in experiments. A HOM dip of nearly 100% was obtained when only the coincidence events from two single photons were postselected [27]. It was also employed to demonstrate the violating Bell inequality using two weak coherent states [28].

In a two-photon interference between a heralded single-photon and a weak coherent state, it is noticed that the spectral purity of the heralded single-photon source plays an essential role in measuring interference visibility [29,30]. In recent years, improving the purity of heralded single-photon sources is extensively studied for the realization of on-chip photon-pair sources [31]. In principle, the visibility of two-photon interference strongly relies on the indistinguishability of two photons. Hence, both spectral and temporal modes for independent sources become important for the performance of two-photon interference. Furthermore, the producing rate in a given time interval is also an important parameter when system noise and the multi-photon event for the input field have to be considered in the experiment. Here, according to the studies of Yuan et al. [32] and Navarrete et al. [33], we show the spectral characterization of two-photon nonclassical interference between a weak coherent state source and another source, which emits a phase-randomized weak coherent state, a single-photon state or a thermal state. A theoretical model with both spectral distribution and average photon number ratio in a time interval was derived. Meanwhile, system noise generated from dark noise and other imperfect correlations is considered. We give the optimum parameters for observing the high-visibility HOM interference. Compared with the reported two-photon interference between independent sources, we focus on establishing a model for a real experiment. Hence, the spectral bandwidth ratio and average photon number ratio are considered in our manuscript. The numerical simulation results can be directly used to analyze experimental data. Furthermore, the bound of two-photon interference visibility is extended by isolating two-photon contributions from the two input states. These numerical results should be of great significance for practical applications such as molecular spectroscopy [34], laser ranging [35], and imaging [36] based on single photon interference. This work may throw some new light on the two-photon interference experiment with independent sources. Furthermore, this paved the way for the high-photon N00N state generated from two-photon or multi-photon interference experiments, demonstrating quantum teleportation and quantum entanglement.

## 2. Theoretical Model

The scheme for our HOM interference measurement consists of two independent input light sources, of which one is a weak coherent state with a varying average photon number, interfering at a BS with another light source with a fixed average photon number. The other source emits a phase-randomized weak coherent state, a thermal state, or a single-photon state. In an experiment, both the thermal and single-photon states can be generated from Type-II SPDC. Either a signal or idler photon can be used as a thermal state; on the other hand, the single photon state can be produced using a heralding method. The setup of our work is shown in Figure 1.

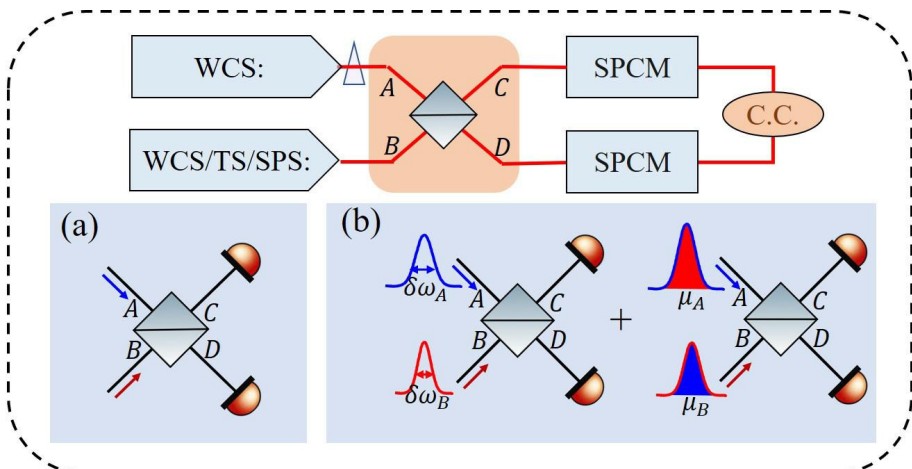

**Figure 1.** HOM interference with different independent sources. (WCS: Weak coherent state; TS: Thermal state; SPS: Single−photon state; $\delta\omega_k(k = A, B)$: spectral width of the source k; $\mu_k(k = A, B)$: average photon number of source k) (**a**) Photons with single mode. (**b**) Photons with spatiotemporal modes.

### 2.1. Single-Mode Photons with an Average Photon Number

The coherent state in a single mode is expanded in photon number state representation, which can be represented as

$$|\alpha\rangle_A = \exp\left(-\frac{|\alpha|^2}{2}\right) \sum_n \frac{\alpha^n}{\sqrt{n!}} |n\rangle, \tag{1}$$

in which $|\alpha|^2 = \mu$ is the average photon number in a given time interval. The $n$ photons probability in that time interval follows a Poisson distribution as $P_{n|\mu} = \mu^n \exp(-\mu)/n!$. On the other hand, the photon number distribution of a single-mode thermal state is called the Bose–Einstein distribution $P_{n|\mu} = \mu^n/(1 + \mu)^{n+1}$ and depends on the average photon number of $\mu$. Meanwhile, the probability of measuring one photon is $\mu$ for a single-photon state with producing rate of $\mu(\mu \leq 1)$ in the given time interval. Hence, when two sources with small average photon number values of $\mu_A$, $\mu_B$ are injected into a BS, the probabilities of finding a pair of photon number states $|m, n\rangle_{AB}$ at input modes are written as follows:

$$P(m_A, n_B | \mu_A, \mu_B)_{\text{coherent}} = \frac{\mu_A^{m_A} \mu_B^{n_B} e^{-\mu_A - \mu_B}}{m_A! n_B!}, \text{for two coherent states} \tag{2}$$

$$P(m_A, n_B | \mu_A, \mu_B)_{\text{thermal}} = \frac{\mu_A^{m_A} \mu_B^{n_B} e^{-\mu_A}}{(1 + \mu_B)^{n_B+1} m_A!}, \text{for coherent state and thermal state} \tag{3}$$

$$P(m_A, 1_B | \mu_A, \mu_B)_{\text{single photon}} = \frac{\mu_A^{m_A} \mu_B e^{-\mu_A}}{m_A!}, \text{for coherent state and single-photon state} \tag{4}$$

where $P(m_A, n_B | \mu_A, \mu_B)$ is the probability of finding the photon number states $|m_A, n_B\rangle$ condition on the input states with average photon number $\mu_A$ and $\mu_B$. The measured coincidence probability at the two outputs is given by

$$\begin{aligned} P^{\mu_A, \mu_B} = {} & P(1_A, 1_B | \mu_A, \mu_B) P(1_C, 1_D | 1_A, 1_B) + P(2_A, 0_B | \mu_A, \mu_B) P(1_C, 1_D | 2_A, 0_B) \\ & + P(0_A, 2_B | \mu_A, \mu_B) P(1_C, 1_D | 0_A, 2_B) + O(m + n > 2), \end{aligned} \tag{5}$$

$P(1_C, 1_D | m_A, n_B)$ is the coincidence probability conditioned on the existence of $m$ and $n$ photons in input ports, and the term $O$ is a positive quantity associated with higher-

order photon terms. Note that, for a coherent state with an average photon number value smaller than 0.22 photons per time interval, the probability for a higher-order photon number (more than 2) is less than 1%. We will keep this restriction in our model. The term $P(1_C, 1_D|1_A, 1_B)$ on the right side of Equation (5) represents the HOM effect and has a value of zero using quantum mechanics predictions when two indistinguishable photons are inputted. Nevertheless, the $P(1_C, 1_D|1_A, 1_B)$ is 0.5 when two distinguishable photons are inputted. The other two terms $P(1_C, 1_D|2_A, 0_B)$ and $P(1_C, 1_D|0_A, 2_B)$ of the right side represent a two-photon in one port and an empty in the other port. Their values are always equal to 0.5. The visibility of the two-photon interference can be defined as

$$V = 1 - \frac{P^{\mu_A, \mu_B}(\text{Indistinguishable photons})}{P^{\mu_A, \mu_B}(\text{Distinguishable photons})}. \tag{6}$$

When two weak coherent states are inputted. It is easy to obtain

$$V_{c-c}(\rho) = \frac{2\mu_A \mu_B}{(\mu_A + \mu_B)^2} = \frac{2\rho}{(1+\rho)^2}, \tag{7}$$

using Equations (2), (5), and (6). Here, $\rho = \mu_A/\mu_B$ is the average photon number ratio of two input states. It indicates that the two-photon interference relates to the average photon number ratio and the maximum two-photon interference visibility occurs at $\rho = 1$ ($\mu_A = \mu_B$), when the high-order terms are ignored. For the interference of coherent state and thermal state, the interference visibility is obtained:

$$V_{c-t}(\rho) \approx \frac{2\mu_A \mu_B}{(\mu_A + \mu_B)^2 + \mu_B^2} = \frac{2\rho}{1 + (1+\rho)^2}, \tag{8}$$

when high-order terms of $\mu_A$ and $\mu_B$ are ignored. For the case of $\rho = \sqrt{2}$, a HOM interference visibility of around 0.4 can be obtained since the Bose–Einstein distribution has almost similar values to the Poisson distribution at a smaller average photon number. Meanwhile, the difference between the two distributions makes the maximum visibility less than 0.5.

The measured coincidence probability for the interference of the coherent state and single-photon state is given $P_{c-s}^{\mu_A, \mu_B} = \mu_A \mu_B P(1_C, 1_D|1_A, 1_B) + (\mu_A^2/2)P(1_C, 1_D|2_A, 0_B)$. Note that the contribution from the two-photon state of the coherent state includes the probability of $P(1_C, 1_D|2_A, 1_B)$. And the interference visibility between the coherent state and single photon state can be written as

$$V_{c-s}(\rho) = \frac{2\mu_B}{\mu_A + 2\mu_B} = \frac{2}{2+\rho} \tag{9}$$

This indicates that a visibility of more than 0.5 always occurs when $\rho < 2$. This is different from the interference results between two coherent states and between a coherent state and a thermal state, in which the interference visibility is less than 0.5. The two photon probability of a coherent state limits the HOM interference. For the case of a perfect single-photon state ($\mu_B = 1$), the interference visibility becomes $2/(\mu_A + 2)$ and reaches near 1 when $\mu_A \ll 1$. For general two-photon interference experiments, system noise plays a very critical role. So we obtain the modified interference visibility of

$$V'_{c-c}(\rho) = \frac{2\rho}{(1+\rho)^2 + \gamma}, \tag{10}$$

$$V'_{c-t}(\rho) = \frac{2\rho}{(1+\rho)^2 + 1 + \gamma}, \tag{11}$$

$$V'_{c-s}(\rho) = \frac{2\rho}{\rho^2 + 2\rho + \gamma}, \tag{12}$$

in an experiment, where $\gamma$ is the system noise parameter, obtained by the ratio of system noise coincidence counts to the coincidence counts of the input state. Figure 2a shows the two-photon interference visibility as a function of the average photon number ratio with different system noise between two independent sources. As we know, the system noise induces the right shift of the maximum visibility. Hence, there is an optimum average photon number ratio to achieve maximum visibility in a real experiment.

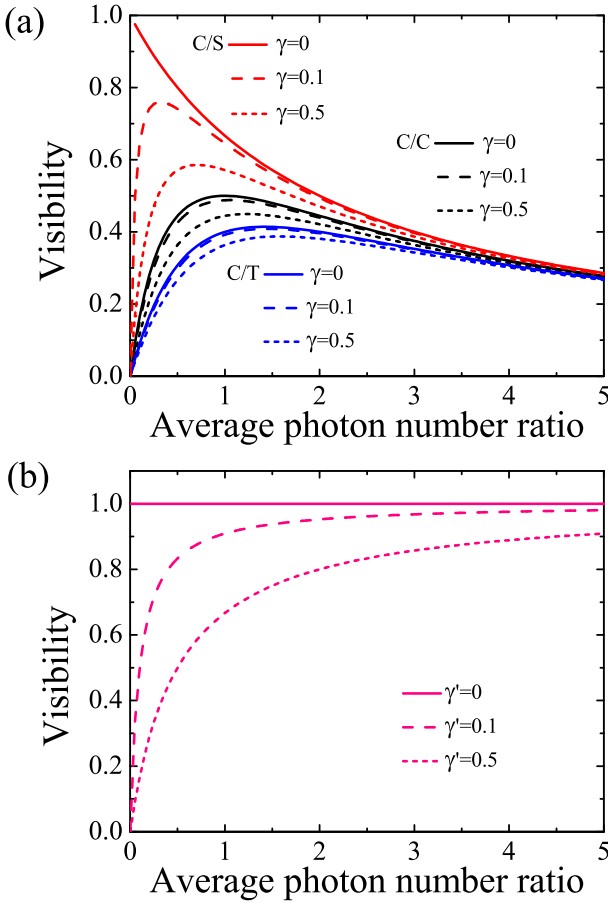

**Figure 2.** Visibility as a function of the average photon number ratio between two independent sources with single mode (**a**) Before isolation two−photon events (**b**) After isolation two−photon events.

The above results show that the reduction of interference visibility is caused by the two-photon contribution of the two input states. Recently, a method was developed to isolate two-photon coincidence [27]. In that case, the interference visibility can be expressed as

$$V''(\rho) = \frac{\rho}{\rho + \gamma'};\qquad(13)$$

where $\gamma'$ is the system noise parameter differing from $\gamma$, obtained by the ratio of system noise coincidence counts to the coincidence counts of $P(1,1|\mu_A, \mu_B)$ of the input state. Since the coincidence of the high-order photon is isolated, a perfect HOM interference occurs. The interference visibility is independent of the average photon number when the system noise is ignored. Otherwise, a visibility of near 1 occurs when $\rho \gg \gamma'$, which means that the system noise is small enough. Figure 2b gives the visibility as a function of the average photon number ratio at different system noise parameters.

*2.2. Photons with Spatiotemporal Modes*

To obtain a deeper understanding, let us consider a description of the photons with spatiotemporal wave packets. For a 50:50 optical BS with spatial modes marked in Figure 1b, the electric field operators can be attributed to its input as

$$\hat{E}_A^+(t) = \xi_A(t)\hat{a}_A, \tag{14}$$

$$\hat{E}_B^+(t) = \xi_B(t)\hat{a}_B. \tag{15}$$

where $\xi_A(t)$ and $\xi_B(t)$, composed of an amplitude envelope and a phase, are envelope wave functions in a time domain and the spatial position has been taken as a subscript. Then, the output electric field of the BS can be obtained by the operators

$$\hat{E}_C^+(t) = \frac{1}{\sqrt{2}}(\hat{E}_A^+(t) + \hat{E}_B^+(t)) = \frac{1}{\sqrt{2}}(\xi_A(t)\hat{a}_A + \xi_B(t)\hat{a}_B), \tag{16}$$

$$\hat{E}_D^+(t) = \frac{1}{\sqrt{2}}(\hat{E}_A^+(t) - \hat{E}_B^+(t)) = \frac{1}{\sqrt{2}}(\xi_A(t)\hat{a}_A - \xi_B(t)\hat{a}_B). \tag{17}$$

To obtain a two-photon coincidence between the two outputs at time $t_1$ and $t_2$, the field operators can be given as $\hat{E}_C^{\pm}(t_1)$ and $\hat{E}_D^{\pm}(t_2)$. With an input state of $\mid \psi_{in}\rangle$, the coincidence probability can be computed with

$$P(t_1, t_2) \propto \langle \psi_{in} \mid \hat{E}_D^-(t_2)\hat{E}_C^-(t_1)\hat{E}_C^+(t_1)E_D^+(t_2) \mid \psi_{in}\rangle. \tag{18}$$

Since the analysis is restricted to two photons, the possible input states of two photons are $\mid \psi_{in}\rangle = \mid 1_A, 1_B\rangle = \hat{a}_A^{\dagger}\hat{a}_B^{\dagger} \mid 0,0\rangle_{A,B}$, $\mid 2_A, 0_B\rangle = (\sqrt{2})^{-1}\hat{a}_A^{\dagger}\hat{a}_A^{\dagger} \mid 0_A, 0_B\rangle$, or $\mid 0_A, 2_B\rangle = (\sqrt{2})^{-1}\hat{a}_B^{\dagger}\hat{a}_B^{\dagger} \mid 0,0\rangle$, and the relative delay between the two input modes of $\tau$ is introduced. Then, the coincident probabilities become,

$$P^{1,1}(t_1, t_2, \tau) = \frac{1}{4} \mid \xi_A(t_1 - \tau)\xi_B(t_2) - \xi_A(t_2 - \tau)\xi_B(t_1) \mid^2, \tag{19}$$

$$P^{2,0}(t_1, t_2, \tau) = \frac{1}{4} \mid \xi_A(t_1 - \tau)\xi_A(t_2 - \tau) \mid^2, \tag{20}$$

$$P^{0,2}(t_1, t_2, \tau) = \frac{1}{4} \mid \xi_B(t_1)\xi_B(t_2) \mid^2, \tag{21}$$

respectively. In a real experiment, we measured the two-photon coincidence probability with some finite time resolution, which is much larger than coincidence time, so it is needed to integrate over all values of $t_1, t_2$. Hence, the coincidence probability recorded in the experiment will be

$$P_c^{m,n}(\tau) = \int dt_1 \int dt_2 P^{m,n}(t_1, t_2, \tau), \tag{22}$$

Hence, the interference visibility can be obtained as

$$V(\tau) = \frac{\sum_{m,n} P_{coinc}^{m,n}(\tau \to \infty) - \sum_{m.n} P_{coinc}^{m,n}(\tau)}{\sum_{m,n} P_{coinc}^{m,n}(\tau \to \infty)}. \tag{23}$$

where, $(m, n) \in \{(1,1), (0,2), (2,0)\}$ indicates the photon number of input state at ports A and B. This gives a two-photon interference behavior depending on the temporal property.

## 3. Interference between Different Sources

A standard way to address the issue of spatiotemporal modes is based on the Fourier transformation, i.e., the decomposition of temporal wave packets into an infinite number

of single-frequency field modes [37–39]. For the input states, they are represented as a superposition of all possible frequency modes occupied by a quantum excitation,

$$|1,1\rangle_{A,B} = \iint d\omega_1 d\omega_2 \phi_A(\omega_1)\phi_B(\omega_2)\hat{a}_A^\dagger(\omega_1)\hat{a}_B^\dagger(\omega_2)|0,0\rangle_{A,B}, \tag{24}$$

$$|2,0\rangle_{A,B} = \frac{1}{\sqrt{2}} \int d\omega_1 \phi_A(\omega_1)\phi_A(\omega_1)(\hat{a}_A^\dagger(\omega_1))^2|0,0\rangle_{A,B}, \tag{25}$$

$$|0,2\rangle_{A,B} = \frac{1}{\sqrt{2}} \int d\omega_2 \phi_B(\omega_2)\phi_B(\omega_2)(\hat{a}_B^\dagger(\omega_2))^2|0,0\rangle_{A,B}, \tag{26}$$

where the spectral amplitude function of the state $\phi_k(\omega)$ is assumed to be normalized so that $\int d\omega \phi^*(\omega)\phi(\omega) = 1$. The spectral amplitude function $\phi_k(\omega)$ and the wave function in the time domain $\xi_k(t)$ have a relationship of

$$\xi_k(t) = \frac{1}{\sqrt{2\pi}} \int d\omega \phi_k(\omega)e^{-i\omega t}. \tag{27}$$

*3.1. Two Weak Coherent States*

Consider two coherent states with Gaussian spectral amplitude function

$$\phi_k(\omega) = \frac{1}{\sqrt[4]{\pi}\sqrt{\delta\omega_k}} \exp\left(-\frac{(\omega - \bar{\omega}_k)^2}{2\delta\omega_k^2}\right), \quad (k = A, B) \tag{28}$$

where $\bar{\omega}_k$ is the central frequency of the field, $\delta\omega_k$ defines its spectral width, and the normalization is taken. From Equation (27), the temporal wave function is

$$\xi_k(t) = \frac{\sqrt{\delta\omega_k}}{\sqrt[4]{\pi}} \exp\left(-\frac{\delta\omega_k^2 t^2}{2}\right) \exp(-i\bar{\omega}_k t). \quad (k = A, B) \tag{29}$$

We now use Equations (19-22) to calculate the coincident probability. The integration can be evaluated using the Mathematica software. The coincidence probability simplifies to,

$$P_{coinc}^{1,1}(\tau) = \frac{1}{2} - \frac{\delta\omega_A \delta\omega_B}{\delta\omega_A^2 + \delta\omega_B^2} \cdot \exp\left(-\frac{\delta\omega_A^2 \delta\omega_B^2}{\delta\omega_A^2 + \delta\omega_B^2}\tau^2\right), \tag{30}$$

$$P_{coinc}^{2,0}(\tau) = P_{coinc}^{0,2}(\tau) = \frac{1}{2}, \tag{31}$$

where the two coherent states with the same central frequency ($\bar{\omega}_A = \bar{\omega}_B$) have been taken. When $\bar{\omega}_A \neq \bar{\omega}_B$, a quantum beat signal with a carrier frequency of $\Delta = |\bar{\omega}_A - \bar{\omega}_B|$ in the interference, will appear. In the following, only the case of the same central frequency is considered. Considering two coherent sources with an average photon number of $\mu_A$ and $\mu_B$, the coincidence probability can be expressed using Equation (5),

$$P_{coinc}^{\mu_A,\mu_B}(\tau) = \frac{(\mu_A + \mu_B)^2}{4} - \mu_A\mu_B \frac{\delta\omega_A \delta\omega_B}{\delta\omega_A^2 + \delta\omega_B^2} \exp\left(-\frac{\delta\omega_A^2 \delta\omega_B^2}{\delta\omega_A^2 + \delta\omega_B^2}\tau^2\right). \tag{32}$$

And the interference visibility is given as,

$$V'(\tau, \rho) = \frac{4\rho}{(1+\rho)^2} \frac{\delta\omega_A \delta\omega_B}{\delta\omega_A^2 + \delta\omega_B^2} \exp\left(-\frac{\delta\omega_A^2 \delta\omega_B^2}{\delta\omega_A^2 + \delta\omega_B^2}\tau^2\right). \tag{33}$$

where the system noise is ignored. This clearly indicates that the maximum visibility ($\tau = 0$) occurring at the two have the same spectral width ($\delta\omega_A = \delta\omega_B$). As is the same with the average photon number ratio, any unequal spectral width will reduce the interference visibility. Meanwhile, the full width of half maximum (FWHM) of the HOM dip depends

on not only the spectral width ratio but also the original spectral width. It is noted that the spectral width ratio is the ratio of FWHM of the spectral width of the two independent sources. The system noise also always reduces the interference visibility.

### 3.2. Weak Coherent State and Thermal State

Usually, the spectral amplitude of a thermal state is the Lorentzian distribution at a center frequency of $\bar{\omega}_B$ and a spectral width of $\delta\omega_B$,

$$\phi_B(\omega) = \frac{1}{\sqrt{2\pi}} \frac{\delta\omega_B}{(\omega - \bar{\omega}_B)^2 + (\delta\omega_B)^2}. \tag{34}$$

This interferes with a weak coherent state with a Gaussian spectral distribution. Their amplitude function in the time domain can be obtained using the Fourier transformation of spectral amplitudes

$$\xi_A(t) = \frac{\sqrt{\delta\omega_A}}{\sqrt[4]{\pi}} \exp\left(-\frac{\delta\omega_A^2 t^2}{2}\right) \exp(-i\bar{\omega}_A t), \tag{35}$$

$$\xi_B(t) = \sqrt{\delta\omega_B} \exp\left(-\delta\omega_B \mid t \mid\right) \exp(-i\bar{\omega}_B t). \tag{36}$$

We calculate and simplify the coincident probability using the above-mentioned method:

$$Pc^{1,1}(\tau) = \frac{1}{2} - \frac{\sqrt{\pi}\delta\omega_B}{4\delta\omega_A} \exp\left(\frac{\delta\omega_B}{\delta\omega_A}\right)^2 \cdot \left[ \exp(2\delta\omega_B\tau) \operatorname{erfc}\left(\frac{\delta\omega_B}{\sqrt{2}\delta\omega_A} + \frac{\delta\omega_A\tau}{\sqrt{2}}\right) \right.$$
$$\left. + \exp(-2\delta\omega_B\tau) \operatorname{erfc}\left(\frac{\delta\omega_B}{\sqrt{2}\delta\omega_A} - \frac{\delta\omega_A\tau}{\sqrt{2}}\right) \right]^2, \tag{37}$$

$$Pc^{2,0}(\tau) = \frac{1}{2}, \ Pc^{0,2}(\tau) = \frac{1}{2}. \tag{38}$$

And the interference visibility can be expressed as

$$V'(\tau,\rho) = \frac{\sqrt{\pi}\rho}{(1+\rho)^2 + 1} \frac{\delta\omega_B}{\delta\omega_A} \exp\left[\left(\frac{\delta\omega_B}{\delta\omega_A}\right)^2\right] \cdot \left[ \exp(2\delta\omega_B\tau) \operatorname{erfc}\left(\frac{\delta\omega_B}{\sqrt{2}\delta\omega_A} + \frac{\delta\omega_A\tau}{\sqrt{2}}\right) \right.$$
$$\left. + \exp(-2\delta\omega_B\tau) \operatorname{erfc}\left(\frac{\delta\omega_B}{\sqrt{2}\delta\omega_A} - \frac{\delta\omega_A\tau}{\sqrt{2}}\right) \right]^2. \tag{39}$$

This gives a maximum value of around 0.4 when two states have the average photon number ratio of $\sqrt{2}$ and the same spectral width when the system noise is not considered. This Lorentzian distribution of the thermal light field makes it different from interference with two Gaussian distributions on the condition of $\tau = 0$. It also indicates that the FWHM of the HOM dip is narrower than that of the interference with two Gaussian spectral distributions.

### 3.3. Single-Photon State and Weak Coherent State

Now we consider two-photon interference between a single-photon state and a weak coherent state. Assuming that they have Gaussian distributions, we directly obtained two-photon interference visibility using the above method:

$$V'(\tau,\rho) = \frac{4}{\rho + 2} \frac{\delta\omega_A \delta\omega_B}{\delta\omega_A^2 + \delta\omega_B^2} \exp\left(-\frac{\delta\omega_A^2 \delta\omega_B^2}{\delta\omega_A^2 + \delta\omega_B^2} \tau^2\right). \tag{40}$$

This indicates that the spectral ratio of the two input states is also an important parameter. The maximum interference visibility occurs only when the spectral width is equal and the coherent average photon number is small enough. For a rough estimate, it is impossible to observe a two-photon interference with visibility of more than 0.5 when the spectral ratio is more than 3.5.

As mentioned above, it is possible to improve the interference visibility by isolating the coincidence probability from the two-photon state of the coherent state. In this case, the interference visibilities for three kinds of input states are given by

$$V_{c-c}''(\tau,\rho) = \frac{2\rho}{\rho+\gamma'} \frac{\delta\omega_A\delta\omega_B}{\delta\omega_A^2+\delta\omega_B^2} \exp(-\frac{\delta\omega_A^2\delta\omega_B^2}{\delta\omega_A^2+\delta\omega_B^2}\tau^2), \tag{41}$$

$$V_{c-t}''(\tau,\rho) = \frac{\sqrt{\pi}\rho}{2\rho+\gamma'} \frac{\delta\omega_B}{\delta\omega_A} \exp[(\frac{\delta\omega_B}{\delta\omega_A})^2] \cdot \left[ \exp(2\delta\omega_B\tau) \right.$$
$$\left. \mathrm{erfc}\left(\frac{\delta\omega_B}{\sqrt{2}\delta\omega_A} + \frac{\delta\omega_A\tau}{\sqrt{2}}\right) + \exp(-2\delta\omega_B\tau)\,\mathrm{erfc}\left(\frac{\delta\omega_B}{\sqrt{2}\delta\omega_A} - \frac{\delta\omega_A\tau}{\sqrt{2}}\right) \right]^2, \tag{42}$$

$$V_{c-s}''(\tau,\rho) = \frac{2\rho}{\rho+\gamma'} \frac{\delta\omega_A\delta\omega_B}{\delta\omega_A^2+\delta\omega_B^2} \exp(-\frac{\delta\omega_A^2\delta\omega_B^2}{\delta\omega_A^2+\delta\omega_B^2}\tau^2), \tag{43}$$

where $\gamma'$ is system noise. The maximum measured visibility depends on the system noise, average photon number ratio, and spectral width ratio of the two input states. However, the FWHM of the HOM dip is sensitive to both the spectral ratio and the spectral width.

## 4. Numerical Results

To facilitate a deeper understanding of our model, we give a numerical calculation based on real experimental parameters. Here, pulse trains with a time duration of ps or fs at a central wavelength of 800 nm are considered. In this case, the linewidth $\delta\lambda(\delta\lambda = \frac{\lambda_0\delta\omega}{2\pi c}$, where $c$ is the speed of light and $\lambda_0$ is the central wavelength), with a value range of several nm, can be realized using an optical interference filter. Figure 3 gives the interference results between two weak coherent states. The normalized two-photon coincidence is plotted in Figure 3a as a function of relative delay $\tau$ for different linewidth when the two weak coherent states have the same average photon number with Equation (32). The observed maximum visibility (at $\tau = 0$) is 0.5 when the two input states have the same spectral width, indicating that it is independent of the absolute spectral width. However, the FWHM of the HOM dip for a broad spectral width is less than that of the narrower spectral width. The interference visibilities, as a function of the spectral width ratio of $\delta\omega_A$ to $\delta\omega_B$, are shown in Figure 3b,c using Equations (33) and (41). Obviously, the interference visibility has a different degree of reduction due to the system noise. By isolating the coincidence of a two-photon state in a coherent state, a visibility of more than 0.5 will be observed, as shown by the solid curve in Figure 3c (red solid curve, black solid and blue solid curves are overlapped). For the case of the same average photon number, the maximum interference visibility occurs on the condition of two weak coherent states with the same spectral width. Otherwise, the system needs to be optimized by considering some practical experimental parameters, such as mismatches in the intensities, the states of polarization of the input signals, and the overall intensity of the input signals. It is also worth mentioning that the unequal average photon number will decrease the two-photon interference visibility.

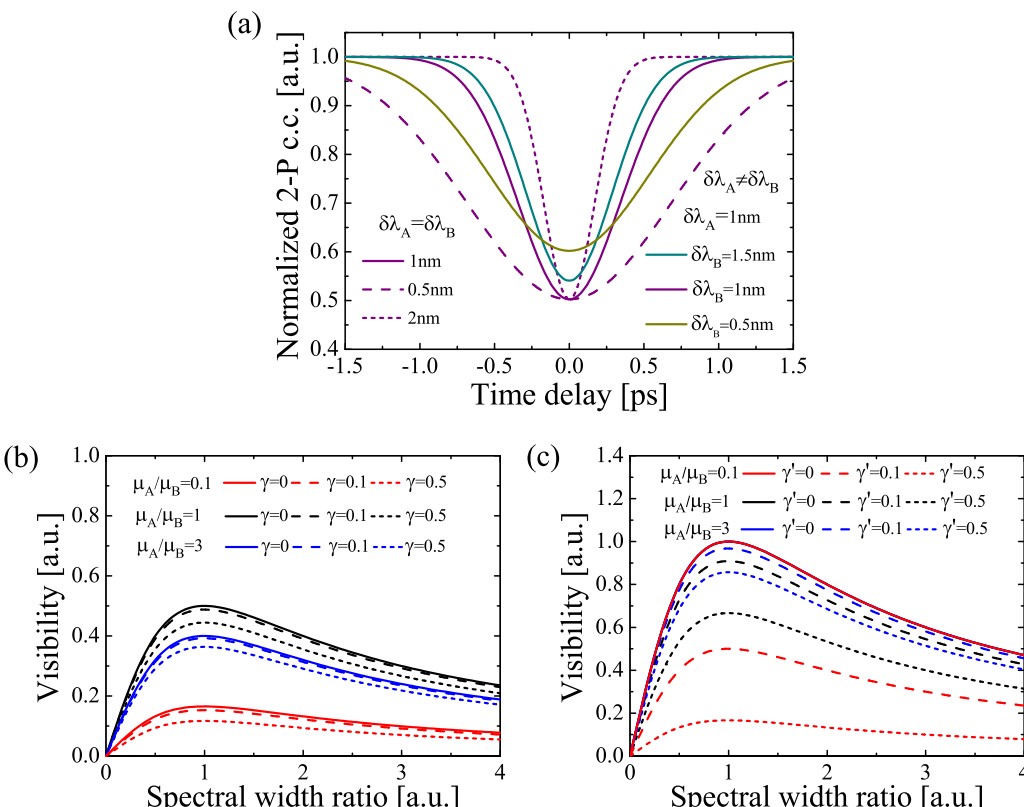

**Figure 3.** Two-photon interference for two weak coherent states. (**a**) Two-photon coincidence as a function of time delay $\tau$; (**b**,**c**) Visibility as a function of spectral width ratio, (**b**) before isolation two−photon events and (**c**) after isolation two−photon events.

Figure 4 shows the two-photon interference between a weak coherent state and a thermal state. In an experiment, it can be realized by interfering laser light with one beam of Type-II SPDC, when another beam is discarded [40]. The result is similar to the two-photon interference between two weak coherent states. However, the best interference visibility of near 0.4 occurs at the average photon number ratio ($\mu_A/\mu_B$) of $\sqrt{2}$ when the coherent state and thermal state have the same spectral width, which is shown in Figure 4b. The Bose–Einstein distribution of photon number leads to a decrease in visibility, compared with the two-photon interference of two-coherent states. This can be understood from Equations (8) and (39). Note that the interference shows very little difference between the Lorentzian and Gaussian distributions of the thermal field for a finite coherence length. By isolating the coincidence of a two-photon state from the coherent state and thermal state, an interference visibility of near 1 is able to be obtained as the same as the interference with two weak coherent states, which is shown in Figure 4c.

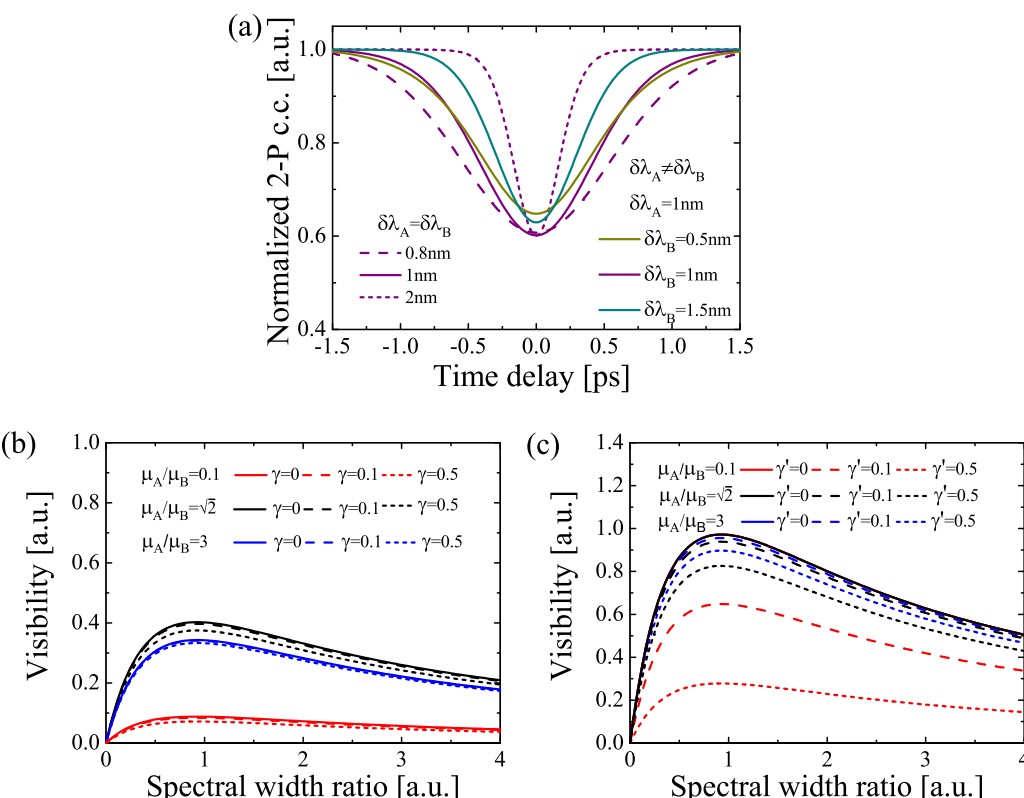

**Figure 4.** Two-photon interference between a weak coherent and a thermal state. (**a**) Two-photon coincidence as a function of time delay $\tau$; (**b,c**) Visibility as a function of spectral width ratio, (**b**) before isolation two-photon events and (**c**) after isolation two$-$photon events.

Figure 5 gives the interference results between a weak coherent state and a single-photon state. Clearly, the two-photon interference with the maximum visibility occurs under the condition of two input states with the same spectral width (Figure 5b). The unequal spectral width will reduce the interference visibility. It is possible to achieve an interference visibility of near 1 (Figure 5b) even without isolating the coincidence of the two-photon state in a coherent state. This can be understood in that the two-photon state probability is much smaller than the one-photon state probability and can be neglected for a weak coherent state. In this case, the system noise becomes important for measuring a high interference visibility (see, also, the red curves in Figure 2a). In Figure 5b, the influence of system noise and coincidence of a two-photon state in a coherent state is shown. The system noise is referred to as the producing rate of a single photon. Hence, a visibility of more than 0.5 is observed when the average photon number of the coherent state is larger than the system noise. By isolating the two-photon coincidence in the coherent state (Figure 5c), the spectral width ratio range for observing interference with visibility of more than 0.5 can be extended. After isolationg the two-photon events in a coherent state, the interference visibility is more dependent on the variation of the system noise for the case of coherent light with low average photon numbers. In a practical experiment, two-photon probability in the coherent state is one of the barriers to improving two-photon interference visibility. Hence, the improvement in producing rate of a single-photon state, or extracting the two-photon probability [27,28], is the key to improving interference visibility. Although experiments on two-photon interference with two single-mode independent sources [21,23,26] or multi-mode coherent states have been extensively investigated, here, we focus more on the effect of unequal spectral width and average photon number ratio on the two-photon interference between the two independents. Meanwhile, the system noise, which plays a significant role in the low average photon interference is also taken into account. This makes the present model most closely related to a practical experiment.

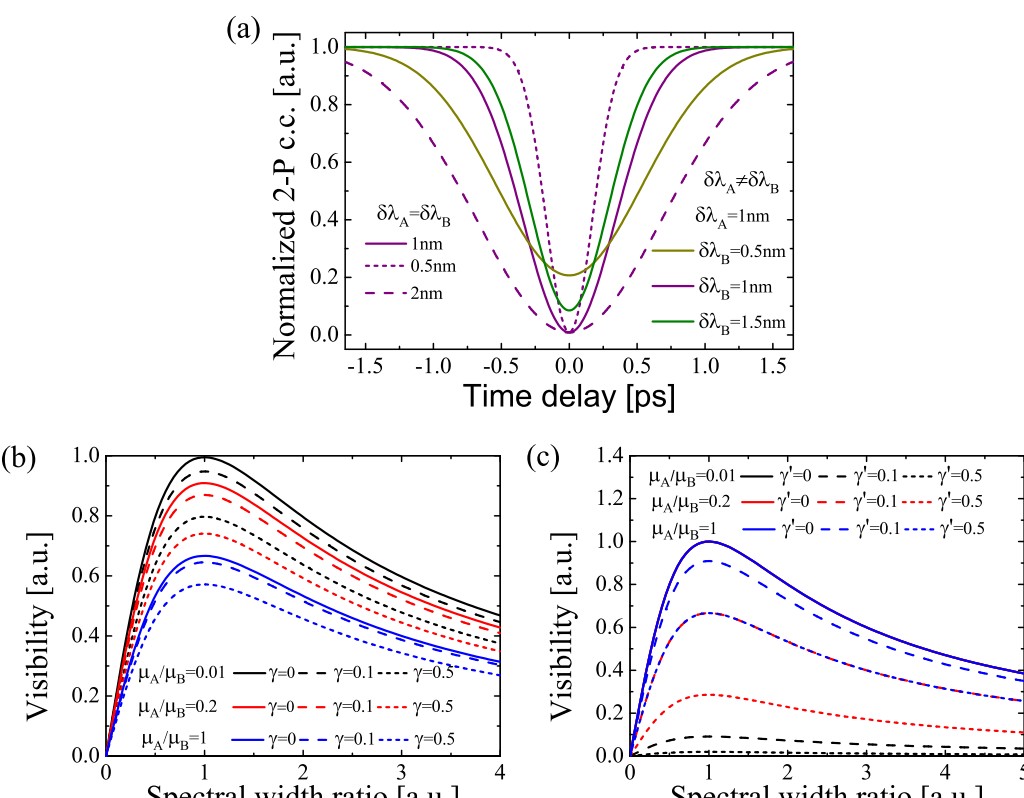

**Figure 5.** Two-photon interference between a weak coherent and a single-photon state. (**a**) Two-photon coincidence as a function of time delay $\tau$; (**b**,**c**) Visibility as a function of spectral width ratio, (**b**) before isolation two-photon events and (**c**) after isolation two-photon events.

## 5. Conclusions

In summary, we present a spectral characterization of two-photon interference between two independent sources. Both analytical and numerical results are given between a weak coherent state source and a phase-randomized weak coherent state, a single-photon state, or a thermal state. Both the spectral ratio and average photon number ratio in a time interval are considered. Furthermore, we considered the effect of system noise on two-photon interference from an experimental viewpoint and achieved a significant improvement in interference visibility by isolating two-photon events from two inputs. The optimum parameters in an experiment are able to be chosen for observing the high visibility Hong–Ou–Mandel interference with independent sources.

**Author Contributions:** Conceptualization, Y.Z.; methodology, L.D. and A.X.; software, L.D.; validation, L.D.; formal analysis, L.D. and Y.Z.; investigation, L.D. and A.X.; data curation, L.D.; writing—original draft preparation, L.D.; writing—review and editing, L.D.; visualization, L.D.; supervision, Y.Z.; project administration, Y.Z.; funding acquisition, Y.Z. All authors have read and agreed to the published version of the manuscript.

**Funding:** This research was funded by JSPS KAKENHI (Grant No. 21K04919, 21K04923).

**Institutional Review Board Statement:** Not applicable.

**Informed Consent Statement:** Not applicable.

**Data Availability Statement:** The data underlying the results presented in this paper are not publicly available at this time but may be obtained from the authors upon reasonable request.

**Conflicts of Interest:** The authors declare no conflict of interest.

## Abbreviations

| | |
|---|---|
| BS | beam splitter |
| SPDC | spontaneous parametric down-conversion |
| HOM | Hong–Ou–Mandel |
| WCS | weak coherent state |
| TS | thermal state |
| SPS | single-photon state |
| FWHM | full width of half maximum |

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
