# Peer review of "Spectral Characterization of Two-Photon Interference between Independent Sources"

_photonics, doi:10.3390/photonics10101125_

Round 1

Reviewer 1 Report

As indicated by the title, the authors demonstrated spectral characterization of two-photon interference between independent sources. Theoretical models for different sources (i.e., two weak coherent states, weak coherent state and thermal state, and single-photon state and weak coherent state, respectively) were given with numerical results. Based on their simulation results, they claimed, e.g., the observed maximum visibility was achieved when the two input states had the same spectral width instead of the absolute spectral width. Such results could be of interest for the practical use of quantum technology. The manuscript may be publishable provided the following issues are properly addressed.

1. The authors should clearly state the purpose and novelty of this work in the introduction part. For instance, the authors mentioned, “Compared with previously reported two-photon interference, both unbalanced spectral width and average photon number ratio in a time interval are considered.” (Line 59) However, why such consideration is important or necessary is unclear. 

2. Line 64, “Furthermore, this paved the way for two-photon or multi-photon interference experiments demonstrating quantum teleportation, and quantum entanglement.” I found this expression was unclear and overstated.

3. Line 196, “Otherwise, the system needs to be optimized.” How and what parameters of the system to be optimized should be stated clearly.

4. Recently, some works using two independent optical frequency combs for single-photon interference measurement for molecular spectroscopy [PNAS 117, 26688 (2020)], laser ranging [IEEE Photonics Technology Letters 33(1), 27 (2021)], and imaging [Nanomaterials 11(6), 1379 (2021)]. I think these works could benefit from the results of this manuscript. Also, these papers could be cited as potential applications of the findings here. 

Reviewer 2 Report

In the quantum optics field, the phenomenon of interference between two single-photon wavepackets at a beam splitter (BS) is recognized as a pivotal aspect. Within this context, the authors of this study present a set of optimal parameters essential for the meticulous observation of high-visibility Hong-Ou-Mandel (HOM) interference. Notably, this investigation distinguishes itself from preceding two-photon interference studies by conscientiously accounting for both the spectral width equilibrium and the average photon number ratios within a specified time interval. Furthermore, the research endeavors to expand the boundaries of two-photon interference visibility by meticulously segregating the two-photon contributions originating from the input states. This work sheds new light on two-photon interference experiments with independent sources and paves the way for demonstrating quantum teleportation and entanglement through multiphoton interference experiments. However, there are some elements that remain unclear and need to be modified. The manuscript should be modified and improved. My suggestions are as follows (see list below).

1. In lines 75-77, the subject of sentence is the two-photon interference patterns. As shown in Fig. 1, Figure 1 describes the interference patterns. Change to “the setup of our work is shown in Fig 1” could be better.

2. In figure 1, the arrow is confused in the areas which are covered by the red or blue. Removing the arrow in blue and red covered areas will make your article more easy to understand.

3. In equations (2a)-(2c), the photon number notation is not consistent with left and right. Please ensure that the notation is consistent on both sides of the equations.

4. In equations (10a) and (10b), the definition of a_A is missing. Furthermore, the notation of equation (11a) and (11b) is missing too. Check the definition and notation to ensure the integrity of your article.

5. The term “spectral width ratio” is mentioned in line 158 first. It is advisable to provide an explicit definition of the spectral width ratio when it is mentioned for the first time.

6. Consistency is important, which will reduce misleading results when reading the article. Figure 3, Figure 4 and Figure 5 lose their consistency because of the inconsistent sequence of curve color. You can use red, black and blue to represent the lowest, mid and highest intensity ratio, respectively like figure 3. Meanwhile, using three different colors instead of red, black and blue to represent the three linewidths is a good choice.

7. In recent years, quantum communication, which leverages the security offered by quantum mechanics, has been gaining significant traction. In this context, the application of two-photon interference is considered an important technique in the measurement device-independent quantum key distribution field. By incorporating pertinent content related to quantum key distribution, the article can further highlight the practical implications and potential applications of the proposed work. Therefore, I suggest that the authors consider citing the following articles [Phys. Rev. Lett. 108, 130503 (2012), Nat. Commun. 5, 3732 (2014), PRX Quantum 3, 020315 (2022), Phys. Rev. Lett. 130, 250801 (2023)] which may make your article be better.

Once the authors have addressed the points raised, I will agree that this work can be published in Photonics.

Minor editing of English language required

Reviewer 3 Report

In this manuscript, authors have presented a spectral characterization of two-photon nonclassical interference on a beam splitter between a weak coherent state source and another source. They also have proposed the possibility to generate multiple emission fields, such as of a phase-randomized weak coherent state, a single-photon state as well as a thermal state.

This method is first proposed for presenting a spectral characterization of two-photon interference between two independent sources. Both analytical and numerical results are given in more detail. The numerical results in this manuscript is interesting, and the novel ideal allows experimental demonstration by making use of current experimental techniques. As a promising and powerful idea, it is a good work and is presented clearly. If there are follow-up experimental results, the feasibility of the experimental program can be further verified.

I did not read the manuscript carefully, just give some obvious problems in manuscript: There's something wrong with the caption font in Fig. 2. Curves overlap words in Fig. 3. The colorful lines of curves are not intuitive and clear, maybe there are better ways to present related image, and so on.

Round 2

Reviewer 1 Report

The authors have modified the manuscript accordingly. I support its acceptance and have no further comments.  

Reviewer 2 Report

This manuscript can be accepted for publication.